



# Aerodynamic model comparison for an X-shaped vertical-axis wind turbine

Adhyanth Giri Ajay[1], Laurence Morgan[2], Yan Wu[1], David Bretos[3], Aurelio Cascales[3], Oscar Pires[3], and Carlos Ferreira[1]

[1]Wind Energy Section, Flow Physics and Technology, Faculty of Aerospace Engineering, Delft University of Technology, Kluyverweg 1, Delft, 2629HS, Netherlands
[2]Dept. of Electronic and Electrical Engineering, University of Strathclyde, Glasgow, G1 1XW, United Kingdom
[3]Wind Energy Department, Centro Nacional de Energías Renovables (CENER), Ciudad de la Innovación, nº 7 · 31621 Sarriguren (Navarra), Spain

**Correspondence:** Adhyanth Giri Ajay (A.GiriAjay@tudelft.nl)

**Abstract.** This article presents a comparison study of different aerodynamic models for an X-shaped vertical-axis wind turbine and offers insight into the 3D aerodynamics of this rotor at fixed pitch offsets. The study compares six different numerical models: a double-multiple streamtube (DMS) model, a 2D actuator cylinder (2DAC) model, an inviscid free-wake vortex model (from CACTUS), a free-wake vortex model with turbulent vorticity (from QBlade), a blade-resolved URANS model, and a Lattice-Boltzmann method (from PowerFLOW). All models, except URANS and PowerFLOW use the same blade element characteristics other than the number of blade elements. This comparison covers the present rotor configuration for several tip-speed ratios and fixed blade pitch offsets without unsteady corrections, except for the URANS and PowerFLOW which cover a single case. The results show that DMS and 2DAC models are inaccurate - especially at highly loaded conditions, are unable to predict the downwind blade vortex interaction, and do not capture the vertical/axial induction this rotor exhibits. The vortex models are consistent with each other and the differences when compared against the URANs and PowerFLOW mostly arise due to the unsteady and flow curvature effects. Furthermore, the influence of vertical induction is very prominent for this rotor and this effect becomes more significant with fixed pitch offsets where the flow at the blade root is considerably altered.

## 1 Introduction

Vertical-axis wind turbines (VAWTs) have attracted significant attention as a promising renewable energy source due to their wind direction independence and their low noise and vibration characteristics (Su et al., 2020). However, the design and optimization of VAWTs pose significant challenges due to their complex aerodynamic characteristics, which are influenced by several factors, including blade geometry, rotor configuration, and wind conditions. To be a viable competitor to horizontal-axis wind turbines (HAWTs), the levelised cost of energy (LCoE) of a single VAWT must be competitive with HAWTs.

The X-Rotor (Leithead et al., 2019) is a novel vertical-axis wind turbine concept that is designed to lower its LCoE for offshore applications. The turbine has two key novel features: an X-shaped primary rotor and the use of secondary tip rotors for power generation (Figure 1). The secondary rotors are attached at the lower blade tips and consequently see a significantly





accelerated inflow speed due to the angular velocity at the primary rotor blade tips. In turn, this allows the secondary rotors to have a very small radius and a large rotational speed. This facilitates the use of cheap, lightweight, high-speed direct drive generators, as opposed to using gearboxes, which significantly reduce the capital costs associated with the turbine. Additionally,

25 the low altitude and mass of the generators eliminate the need for jack-up vessels for maintenance, potentially significantly reducing the associated operations and maintenance costs (Flannigan et al., 2022). The primary rotor is designed to increase the tip-speed ratio and swept area of the rotor (compared to a traditional H-shaped VAWT for the same material used) while cancelling the overturning moments associated with V-shaped VAWT rotors (Kolios et al., 2013; Shires, 2013). The upper blades of the X-Rotor are pitch controlled and are designed to shed aerodynamic power in above-rated conditions. The lower

30 blades are not pitch controlled as any change in it would disrupt the operation of the secondary rotors. A recent study on the operations expenditure of the X-Rotor concept by Flannigan et al. (2022) demonstrated large savings on the operational cost of energy compared to a HAWT. A similar feasibility study by Leithead et al. (2019), showed up to 26% overall cost savings compared to HAWTs. The development of the X-Rotor concept is currently the subject of a European Union Horizon2020 project XROTOR.

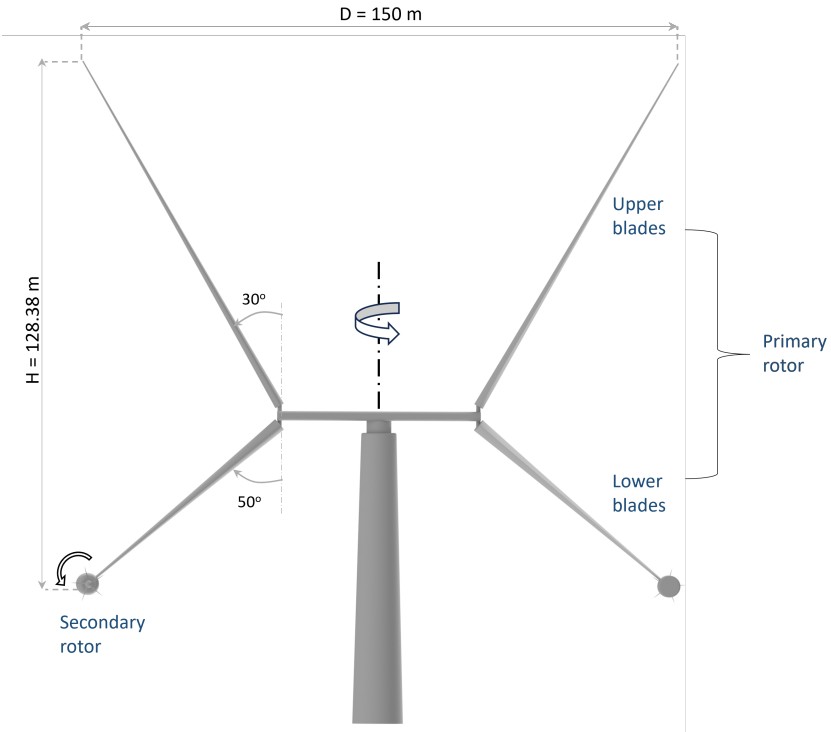

**Figure 1.** A render of the X-Rotor turbine with geometrical dimensions from Leithead et al. (2019).

35 The characterisation of the aerodynamics of the primary rotor is a critical challenge in the design of the X-Rotor turbine. An initial characterisation was completed by Morgan and Leithead (2022), where a double multiple streamtube (DMS) simulation



tool for the X-Rotor was developed and validated against free vortex wake codes. Although this work presented the potential for the DMS tool to characterise the X-Rotor, it was limited to looking at the turbine without pitch offset.

Accurate prediction of VAWT aerodynamic performance is essential for effective design and optimization, as it directly affects the power output and efficiency of the turbine. Different aerodynamic models for VAWTs have been researched and used, each with advantages and limitations. Ferreira et al. (2014) presented detailed blind comparisons between different aerodynamic models for a case study of an H-type VAWT with multiple pitch offsets, concluding that streamtube models behave fundamentally different when pitch offsets are introduced. As VAWTs have finite blade lengths, a spanwise distribution of circulation arises (that varies azimuthally) that in turn leads to spanwise induction and load variations (3D aerodynamic effects). De Tavernier et al. (2020b) showed the effect of 3D aerodynamics (tip and trailing vortices as well) between models for an H-type VAWT at different aspect ratios and identified that the assumptions used in these models limit the 3D aerodynamic behaviour. Keijer (2020), Ferreira (2009), and De Tavernier et al. (2020a) highlighted the importance of accounting for these 3D aerodynamic effects by observing the induction on the wake by the tip-vortices at different aspect ratios, as well as describing the loss in power compared to 2D assumptions. Additionally, it was also concluded that modification of the load distribution of a VAWT may be achieved by fixed blade pitch offsets. Ferreira (2009) investigated the induced three-dimensionality of the tip-vortices on the near wake by varying the pitching axis and observed that the trailing vortex significantly affects performance compared to 2D models. Franchina et al. (2019) conducted a 3D-CFD analysis of an H-type VAWT to obtain the performance at design and off-design conditions and concluded that these 3D effects significantly affect the rotor loads. It also stated that the turbulence models produce more accurate results at lower tip-speed ratios. Analysis of blade-vortex interaction (BVI) and its effect on the fluid flow as the blade passes through its own wake from the previous passage showed this effect was significant even for rotor-level loads (Kozak et al., 2014; Posa and Balaras, 2018).

With the unique configuration of the X-Rotor, there is a significant influence of induction in the vertical direction expected due to the coned blades of this geometry configuration. Therefore, to characterise the aerodynamics of the X-Rotor, it is necessary to understand the discrepancies between different aerodynamic models. In the aforementioned publications, the 3D aerodynamic effects of pitch offsets have been studied in detail but they do not cover that of VAWTs with coned blades, specifically for the X-Rotor.

Hence, the scope of this article is two-fold: (1) present a comparative study of different aerodynamic models for the X-Rotor's primary rotor and (2) investigate the 3D aerodynamics of the X-Rotor associated with fixed blade pitch offsets. The results of this study will provide valuable insights into the development of accurate and efficient VAWT design tools and contribute to the advancement of renewable energy technologies.

The specific objectives of this paper are as follows:

1. Understand the agreement between the models presented here, based on the power, thrust, and blade forces.

2. Obtain the range of operating conditions over which this agreement holds (of particular interest is the pitch offset).

3. Look at the effects of vertical induction to understand where 3D models are necessary for coned VAWT simulations.





## 2 Methodology

### 2.1 Aerodynamic models

The aerodynamic models used in this study are described here. A short summary of the models used along with their fidelity is presented in Table 1. The fidelity is based on the underlying physics of the model. Using a momentum-based approach is considered low-fidelity, a free vortex wake model is considered mid-fidelity and a viscous CFD approach is referred to as high-fidelity. The low- and mid-fidelity models used the airfoil polars that are discussed in Section 2.2.

#### 2.1.1 Double multiple streamtube model (DMS)

The double multiple streamtube model, developed by Paraschivoiu (1981), is a 1D momentum-based model which is expanded to model 3D rotors through a double discretisation scheme. Experimental validation studies for this model were conducted by Paraschivoiu (1982). The 3D rotor is first decomposed into 2D slices along its height, then each slice is split into parallel streamtubes that cross the rotor circumference twice (Figure 2). Blade element momentum theory is first used to solve the flow at the upwind crossing point, then at the downwind crossing point assuming that atmospheric pressure has been recovered. Conservation of mass is assured by allowing streamlines to expand through the method proposed by Sharpe and Freris (1990). The implementation is identical to that described by Morgan and Leithead (2022), however, no unsteady corrections are applied. Tip and root loss corrections from Prandtl et al. (1927) are applied. The rotor is discretised into 120 slices, with each slice discretised into 31 streamtubes (62 azimuthal positions) after a grid convergence study. The formulation of streamtube expansion means that there is no fixed azimuthal discretion as the distance between streamlines is dictated by the loading on the rotor.

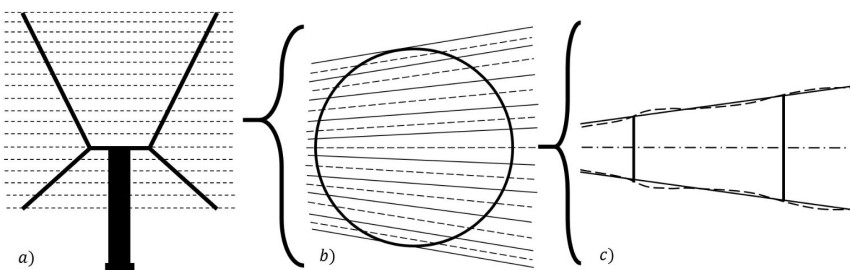

**Figure 2.** The discretisation procedure of the X-Rotor in DMS from Morgan and Leithead (2022). Vertical discretisations are shown in (a). Each vertical plane discretised into streamtubes (solid) and its corresponding central streamline (dashed) is shown in (b). Streamtubes crossing the upwind and downwind actuators are shown in (c).





### 2.1.2 2D actuator cylinder model (2DAC)

The 2D actuator cylinder model, developed by Madsen (1982), is a 2D momentum model that uses the actuator disk concept
for the swept area of a VAWT. The model is based on the 2D Euler equations and a linearised solution (Madsen et al., 2014;
Cheng et al., 2016) is used in this model. In the implementation of an X-shaped VAWT, the 2DAC is used by decomposing
the X-Rotor into 2D slices along its height. Each slice is aerodynamically independent of the other, i.e., it does not account for
induction in the axial/vertical orientation. Tip-loss correction (Prandtl et al., 1927) is introduced at the blade tips. A total of 139
slices of the rotor along the rotation axis are considered for the simulations with each slice containing azimuthal discretisation
of 5°. Each blade section contained 3 slices, with higher refinement of 8 slices at the tips and root. A representation of this
axial discretisation is shown in Figure 3.

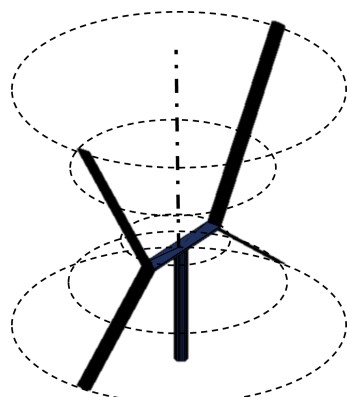

**Figure 3.** Vertical discretisation procedure of the X-Rotor inside the 2DAC. These are constrained to the radius of the rotor at the specific
height of the vertical 2D segment.

### 2.1.3 CACTUS free-wake vortex model (CACTUS)

The Code for Axial and Cross-flow TUrbine Simulation (CACTUS) developed by Murray and Barone (2011), is a three-
dimensional vortex modelling tool for wind turbines. The flowfield is constructed using a vortex lattice, where the velocity
is an arithmetic sum of the freestream and the velocity induced by the vortices. This is calculated through the Biot-Savart
law (Katz and Plotkin, 2009). To simulate the X-Rotor, a free-wake algorithm that calculates the wake convection velocity at
each time step is employed. The upper and lower blades are discretised into 18 blade sections each (minimum to attain blade
element independence of power), with an additional blade section to smoothen the geometry from the upper to the lower blades
with fixed pitch offsets. A constant vortex core model is used and the vortex core is 100% of the chord-to-radius ratio. The
simulations are run for 12 revolutions to attain convergence (discussed in Section 2.2) with a second-order predictor explicit
time advancement scheme.



### 2.1.4 QBlade free-wake vortex model (QBlade)

QBlade is a turbine design and simulation tool to perform aerodynamic studies to facilitate the design of wind turbines developed by Marten et al. (2013). Later updates added lifting-line simulation capabilities (Marten, 2020), that used a free-wake
vortex model with varied vortex core radius. The implementation applied here uses sizes of 100% and 20% chord-to-radius ratio for the bound and trailing vortices, respectively, and a first-order forward integration scheme is used for the wake nodes. The blades are discredited into 20 blade sections for both upper and lower blades (based on the blade element convergence of power). A vortex expansion rate is governed by a turbulence vortex viscosity factor of 2560, which is the default setting in QBlade. The model also used vortex stretching, with the maximum value of the stretching factor of $1 \times 10^6$. The blade
discretisation and vortex lattice system from this simulation is presented in Figure 4.

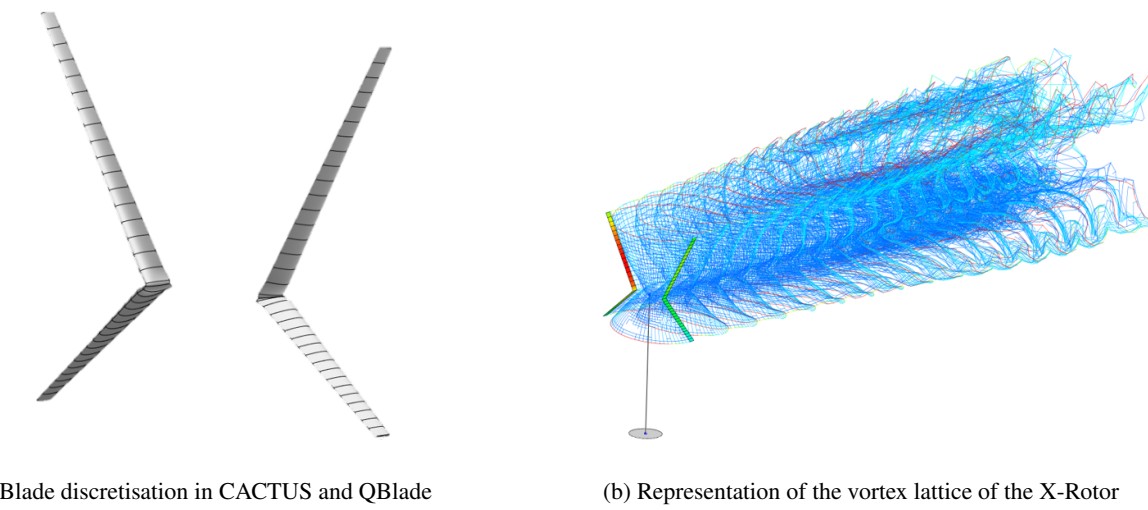

(a) Blade discretisation in CACTUS and QBlade                    (b) Representation of the vortex lattice of the X-Rotor

**Figure 4.** Discretisation and vortex lattice for CACTUS and QBlade

### 2.1.5 Unsteady Reynolds Averaged Navier-Stokes (URANS) CFD model

The X-Rotor is also simulated using a URANS model through OpenFOAM (Weller et al., 1998), an open-source finite-volume CFD tool used for fluid dynamic simulations. The fluid domain is modelled with a blade-resolved mesh with the $k - \omega$ shear-stress transport (SST) turbulence closure model (Menter, 1993) and used the PIMPLE scheme for the pressure-velocity cou-
pling. The implementation here uses 72 million cells (grid-independence attained) with domain lengths of $8.26D$, $5.33D$, and $4D$ in the streamwise, lateral, and axial directions, where $D$ is the primary rotor diameter. Although previously VAWT CFD simulations with larger domain sizes of $20D$ were considered as best practices (Shamsoddin and Porté-Agel, 2014; Rezaeiha et al., 2017a; Belabes and Paraschivoiu, 2023), these domain sizes are not considered here as the focus was not on flowfield and wake. However, this could potentially impact the rotor thrust, which is discussed in Section 3.1.2. Mesh refinements are
performed through snappyHexMesh with the detailed information provided in Appendix B.





### 2.1.6 PowerFLOW (PFLOW)

PowerFLOW 6-2021-R2, developed by 3DS SIMULIA (Dassault Systemes, 2021), solves the discrete, explicit, transient, compressible Lattice-Boltzmann method (LBM) by statistically tracking the streaming and collisions of fluid particles for a finite number of directions. A more detailed description of this method is presented in the work of Shan et al. (2006). A Very

Large Eddy Simulation (VLES) model is implemented to account for the effect of the sub-grid unresolved scales of turbulence. A two-equation $k - \epsilon$ re-normalization group is used to compute a turbulent relaxation time added to the viscous relaxation time. This model simulated the coupled primary and secondary rotor, including the tower and cross-beam (strut), unlike the other models in this study. The original purpose of this model was to understand the aero-acoustic effects of the secondary rotor, which is outside the scope of this study and only the primary rotor's loads and performance are considered here. In

order to reduce the computational cost, a pressure-gradient-extended wall model is used to approximate the no-slip boundary condition on solid walls. This implementation used 73.45 million cells and 7.7 million surface elements with domain lengths of $27.75D$, $37D$, and $5.72D$ in the stream, lateral, and axial directions. Further information is provided in Appendix C.

**Table 1.** Models considered for benchmarking

| Model | Method | Fidelity (colour in plot) |
|---|---|---|
| DMS | Double multiple streamtube | Low (orange) |
| 2DAC | 2D Actuator Cylinder + Blade element momentum (BEM) | Low (orange) |
| QBlade | QBlade Lifting line free vortex wake | Medium (blue) |
| CACTUS | CACTUS Lifting line free vortex wake | Medium (blue) |
| URANS | Unsteady Reynolds Averaged Navier-Stokes OpenFOAM | High (black) |
| PFLOW | Lattice Boltzmann VLES | High black) |

### 2.2 Test setup

The X-Rotor rotor geometry consists of a radius of $25\,\mathrm{m}$ and $75\,\mathrm{m}$ at the root and the tip of the blades respectively. The upper

blades are attached at $30°$ from the vertical plane and therefore have a length of $100\,\mathrm{m}$, and the lower blades are connected at $50°$ from the vertical plane and therefore have a length of $65.3\,\mathrm{m}$. Both sets of blades have a linear taper in chord and relative thickness and utilise the symmetric NACA00XX aerofoil family. The blades are untwisted. The blade geometry at the root and tip are given in Table 2. For all simulation tools that are not blade resolved, polars are generated for the airfoil profile range of NACA0008 (root section) to NACA0025 (tip section) using XFOIL (Drela, 1989) at Reynolds' number of $Re = 1.5 \times 10^7$

(based on the chord at the tip), and then extrapolated through the Viterna and Janetzke (1982) method. In all models (except PFLOW) only the aerodynamically active portion of the rotor was modelled, meaning that the cross-beam connecting the rotor blades, and the tower are not modelled.

Due to the coned blades, the local tip-speed ratio pertaining to the local blade elements varies along the span and height. Therefore, the tip-speed ratios considered for analysis represent the value at the blade tips. Each of the low-fidelity and mid-



**Table 2.** The geometry for the X-Rotor blades, and intermediate values can be found through linear interpolation.

| Blades | Section | Radius [m] | Cone Angle [°] | Chord [m] | Twist [°] | Thickness [%] |
|--------|---------|------------|----------------|-----------|-----------|---------------|
| Upper  | tip     | 75         | 30             | 5         | 0         | 08            |
|        | root    | 25         | 30             | 10        | 0         | 25            |
| Lower  | tip     | 75         | 50             | 7         | 0         | 08            |
|        | root    | 25         | 50             | 14        | 0         | 25            |

fidelity models simulated the aerodynamic performance and loads of the turbine at a tip-speed ratio range of $\lambda = [2.5, 5]$ at 0.5 intervals with a fixed upper blade pitch range of $\beta = [-20°, 20°]$ with $5°$ intervals. Positive pitch corresponds to the upper blades pitching into the axis of rotation and negative pitch corresponds to the blades pitching away from the rotation axis. The pitching axis position varies from 25% chord at the root and 50% chord at the tips (see Appendix A). Figure 5 shows the difference between the airfoil orientation between positive and negative pitch offsets compared to no pitch offset.

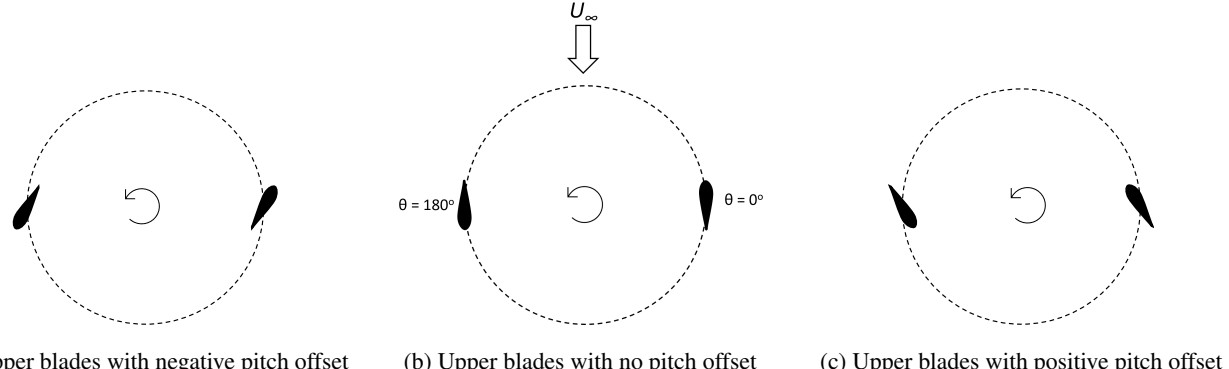

(a) Upper blades with negative pitch offset     (b) Upper blades with no pitch offset     (c) Upper blades with positive pitch offset

**Figure 5.** Top view of upper blade pitch offsets. $\theta$ indicates the azimuth and $U_\infty$ is the freestream velocity.

A homogeneous, constant, freestream velocity of $U_\infty = 12\,\mathrm{m\,s^{-1}}$ is used for all low and mid-fidelity models. The 2DAC, CACTUS, and QBlade have 72 azimuthal discretisations per rotation, but this is not enforced in the DMS model, as the streamtube discretisation does not allow for uniform azimuthal discretisation. The high-fidelity models used $\lambda = 5$ at $\beta = 0°$ as their test cases with an inlet velocity of $12.5\,\mathrm{m\,s^{-1}}$. Moreover, these models discretised the domain spatially and temporally. The momentum and vortex models are run iteratively to reach a convergence criterion of $1 \times 10^{-3}$ based on the power. In 160   the vortex models, the models completed 12 revolutions to reach this value. The convergence criterion value is chosen to give optimal accuracy for computational time, as a lower value becomes dependent on the interpolation scheme each model used to obtain the angle of attack from the polars. Therefore, the vortex model results presented here are from the last revolution of the rotor.

     While the effects of flow curvature (Rainbird et al., 2015; Migliore et al., 1980; Balduzzi et al., 2014) and dynamic stall 165   (Masson et al., 1998; Le Fouest and Mulleners, 2022) are shown to significantly affect aerodynamic performance, the low- and





mid-fidelity models have been implemented without these effects. The choice to omit them is made in order to compare the underlying flow models as, at present, these codes do not employ identical correction factors for dynamic stall or flow curvature effects. Any one of these models could be updated with identical correction factors, however, a comparison of the most valid correction factors is beyond the scope of this article.

Both high-fidelity models inherently include flow curvature, dynamic stall, and viscous terms, as the rotor in these models is blade-resolved. Additionally, the PFLOW simulations include the secondary rotor in the lower blade, the tower, and the cross-beam that connects the two sets of blades together. As the secondary rotors experience high thrust, this is expected to decrease the tangential forces of the lower blade in the azimuth where the blade does not experience stall, compared to the other simulation cases. As the cross-beam is not as aerodynamically significant compared to the blades of the rotor, the

aerodynamic effects from this surface are neglected. The lower blade loads are also expected to be affected by the wake of the tower. However, this is expected to only have a significant impact around an azimuth $\theta = 180°$.

## 3    Results and discussions

### 3.1    Study of rotor power and thrust

The results for power and thrust of the X-Rotor are discussed in detail in Section 3.1.1 and Section 3.1.2. They are represented

in terms of non-dimensional coefficients given by:

$$C_P = \frac{P}{\frac{1}{2}\rho A U_\infty^3}, \tag{1}$$

$$C_T = \frac{T}{\frac{1}{2}\rho A U_\infty^2}, \tag{2}$$

where $P$ and $T$ are the revolution-averaged power and thrust, $A$ is the frontal area of the primary rotor ($12\,870\,\mathrm{m}^2$), $U_\infty$ is the

free-stream velocity, and $\rho$ is the density of air, which is $1.225\,\mathrm{kg\,m^{-3}}$.

#### 3.1.1    Power

The variation of $C_P$ with pitch angle $\beta$ in the range of tip-speed ratios $\lambda$ is presented in Figure 6. In these plots, it is observed that DMS severely deviates at $\beta = -10°$ and for all negative pitch offsets from the other models. For positive pitch offsets, the model predicts different values, except at $\beta = 15°$. This is due to the DMS model's assumption that the downwind actuator is

in the fully expanded wake of the upwind actuator, which is not the case with pitch offsets. This shows that DMS is unreliable to evaluate the X-Rotor's aerodynamic performance, which is also observed by Ferreira et al. (2014) for an H-type VAWT. Although it predicts similar values to the other models without pitch offsets, the model fails to capture the aerodynamic power profile of the X-Rotor.

     The mid-fidelity models agree well with each other in pitch ranges of $-10° \leq \beta \leq 10$. The 2DAC model agrees with the

CACTUS results for positive pitch offsets and agrees better with the QBlade results for negative pitch offsets. Compared to the





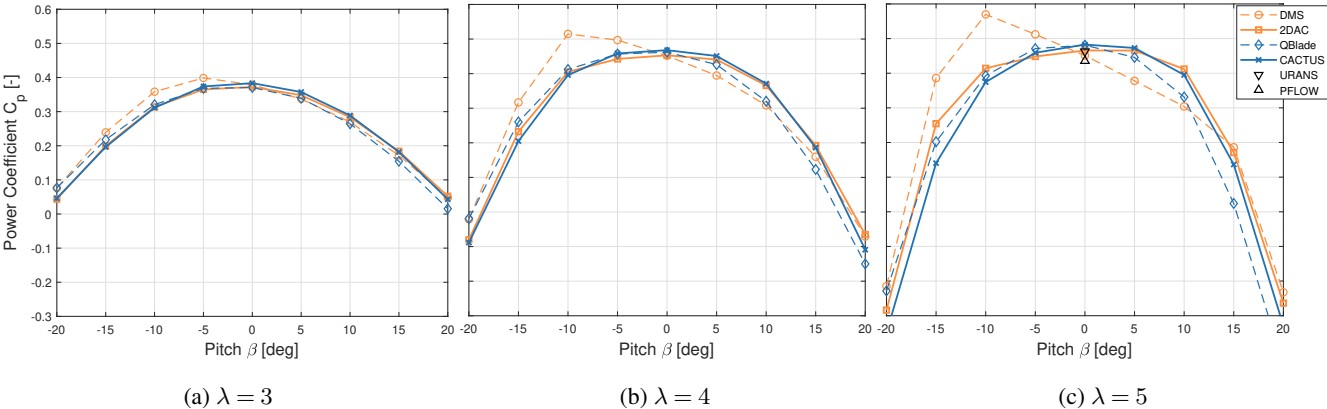

| (a) $\lambda = 3$ | (b) $\lambda = 4$ | (c) $\lambda = 5$ |

**Figure 6.** Variation in rotor power coefficient with pitch angle at fixed tip speed ratios. Orange and blue correspond to the momentum and vortex models respectively. URANS and PowerFLOW results are also shown at $\lambda = 5$ for $\beta = 0°$, corresponding with the black markers.

URANS and PFLOW at $\lambda = 5$, the vortex models over-predict the power slightly (roughly 6%) in contrast to the momentum models agreeing well. For URANS, this is due to the domain size causing blockage effects that results in these differences, which is based off the work by Rezaeiha et al. (2017a), where the correlation between CP and the distance of inlet and outlet from the rotor is tested. The results observed here agree quite well with the correlation. PFLOW shows the least power, but this still is quite close to the other models despite modelling the secondary rotors and the tower. Rezaeiha et al. (2017b) demonstrated the power loss correlation with the dimensions of the tower, which explains the power deficit of the PFLOW observed here.

### 3.1.2 Thrust

The variation of $C_T$ with pitch angle $\beta$ and tip-speed ratio $\lambda$ is shown in Figure 7. While the trends are preserved between models, the magnitudes differ significantly. At $\lambda = 3$ for $\beta \geq 0°$, 2DAC and DMS match quite well with QBlade, until $\beta = 15°$, while CACTUS predicts higher thrust. However at $\beta \leq 0°$, QBlade moves away from the momentum models and agrees well with CACTUS. At $\lambda = 4$, all models predict different thrusts, with the momentum models systematically showing lower thrust with increasing $\lambda$ compared to others. At $\beta \leq 0°$, 2DAC and DMS agree well with each other except at large pitch offsets. This observation is enhanced at $\lambda = 5$, where the CACTUS and QBlade results estimate larger thrust values than the low-fidelity models. However, URANS and PFLOW agree quite well with the vortex models at $\beta = 0°$. The DMS predicts low values due to the limitation of its streamtube assumption, as mentioned earlier. Additionally, it can be said that at high loading, the model fails to show accurate behaviour, as its profile looks entirely different from the others. 2DAC's low thrust prediction can be attributed to the limitation of the linear correction method, where it becomes less accurate at higher loading (Ferreira et al., 2014). The difference between the URANS and the vortex models (around 2%) is due to the domain size chosen for the simulation in URANS (Rezaeiha et al., 2017a). All models deviate significantly at large pitch offsets, which in a way exaggerates the differences observed at small pitch offsets.





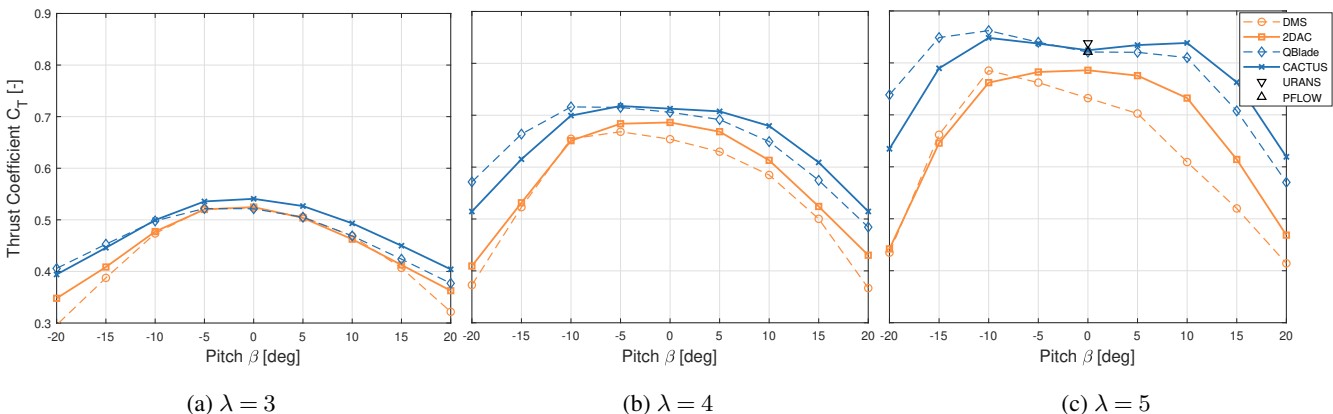

(a) $\lambda = 3$          (b) $\lambda = 4$          (c) $\lambda = 5$

**Figure 7.** Variation in rotor thrust coefficient with pitch angle at fixed tip speed ratio. Orange and blue correspond to the momentum and vortex models respectively. URANS and PowerFLOW results are shown at $\lambda = 5$ for $\beta = 0°$, corresponding with the black markers.

Overall, the vortex models seem to agree well with the high-fidelity models, suggesting the momentum models are not quite valid at these high tip-speed ratios.

## 3.2 Study of blade forces

As the URANS and PFLOW predicted the results for one case, the normal and tangential loads are compared with the momentum and the vortex models in Section 3.2.1. The momentum and vortex models are analysed in detail for the set pitch offset cases in Section 3.2.2.

### 3.2.1 Blade forces - comparison with high-fidelity models

The normal and tangential forces at $\lambda = 5$ and $\beta = 0°$ for the upper and lower blades from all the models are shown in Figure 8
and Figure 9, respectively. The upper blade normal forces of QBlade and the high-fidelity models match well in the upwind half, but the latter predict lower magnitudes in the downwind half. Additionally, in the lower blade, URANS systematically predicts higher forces than the other models while PFLOW indicates the fluctuation at $\theta = 270°$ which is due to tower wake interaction. The upper blade tangential forces match quite well in the upwind half. But in the downwind half, URANS and PFLOW predict lower forces compared to the other models. This is also observed in the lower blades, except that PFLOW
predicted lower forces in the upwind half as well. The difference between high-fidelity and other models in both normal and tangential forces is probably due to high flow separation occurring in the downwind half of the turbine. However, isolating its effect from the inherent flow curvature in the high-fidelity model is outside the scope of this paper.





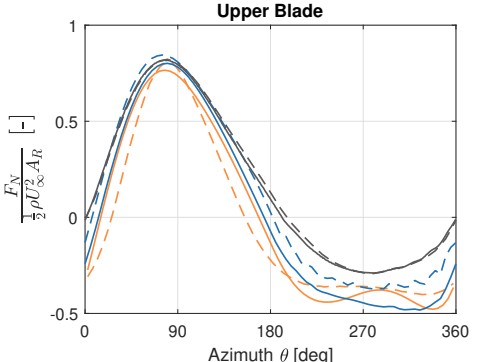
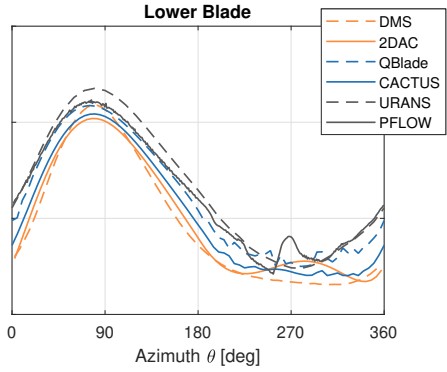

**Figure 8.** Normal forces at $\lambda = 5$ and $\beta = 0°$ predicted by the momentum, vortex, and CFD models. Orange, blue, and grey represent momentum, vortex, and CFD models respectively.

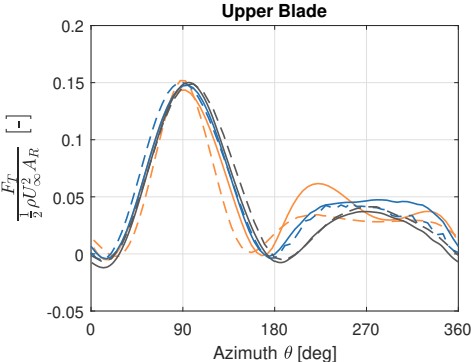
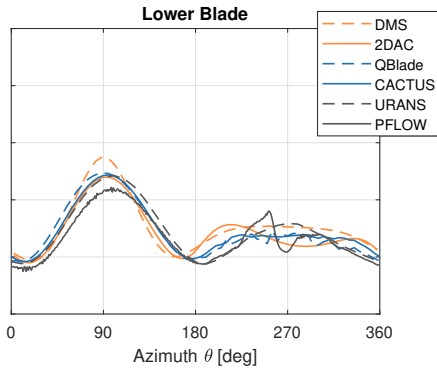

**Figure 9.** Tangential forces at $\lambda = 5$ and $\beta = 0°$ predicted by the momentum, vortex, and CFD models. Orange, blue, and grey represent momentum, vortex, and CFD models respectively.

### 3.2.2 Blade forces - comparison between different pitch-offsets

By increasing $\lambda$, the loads on the blade increase, which is inferred from Figure 7. The results discussed henceforth are limited

to $\lambda = 4$. Pitch offset cases of $\beta = -10°, 0°, 10°$ are chosen for further analysis. The forces of the turbine are calculated by integrating individual blade-element normal force contributions along the span of the blade. The normal forces for upper and lower blades are shown in Figure 10 and the tangential forces are shown in Figure 11.

From the normal force data presented in Figure 10, at $\beta = 0°$, the rotor is loaded more in the upwind half ($\theta = 0°, 180°$) than the downwind half ($\theta = 180°, 360°$). This is due to the VAWT experiencing asymmetric force distribution during operation, as

demonstrated for an H-VAWT (Madsen et al., 2014; Massie et al., 2019). Here, all models agree well except for DMS which slightly under-predicts the forces in the upwind half, although it matches the peak force predicted by the other models. For negative pitch offsets, the force magnitude decreases at the upwind half of the rotor and increases at the downwind half of the rotor, whereas the opposite occurs when the blades have a positive pitch offset. At $\beta = -10°$, the models significantly deviate



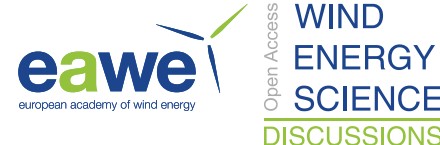

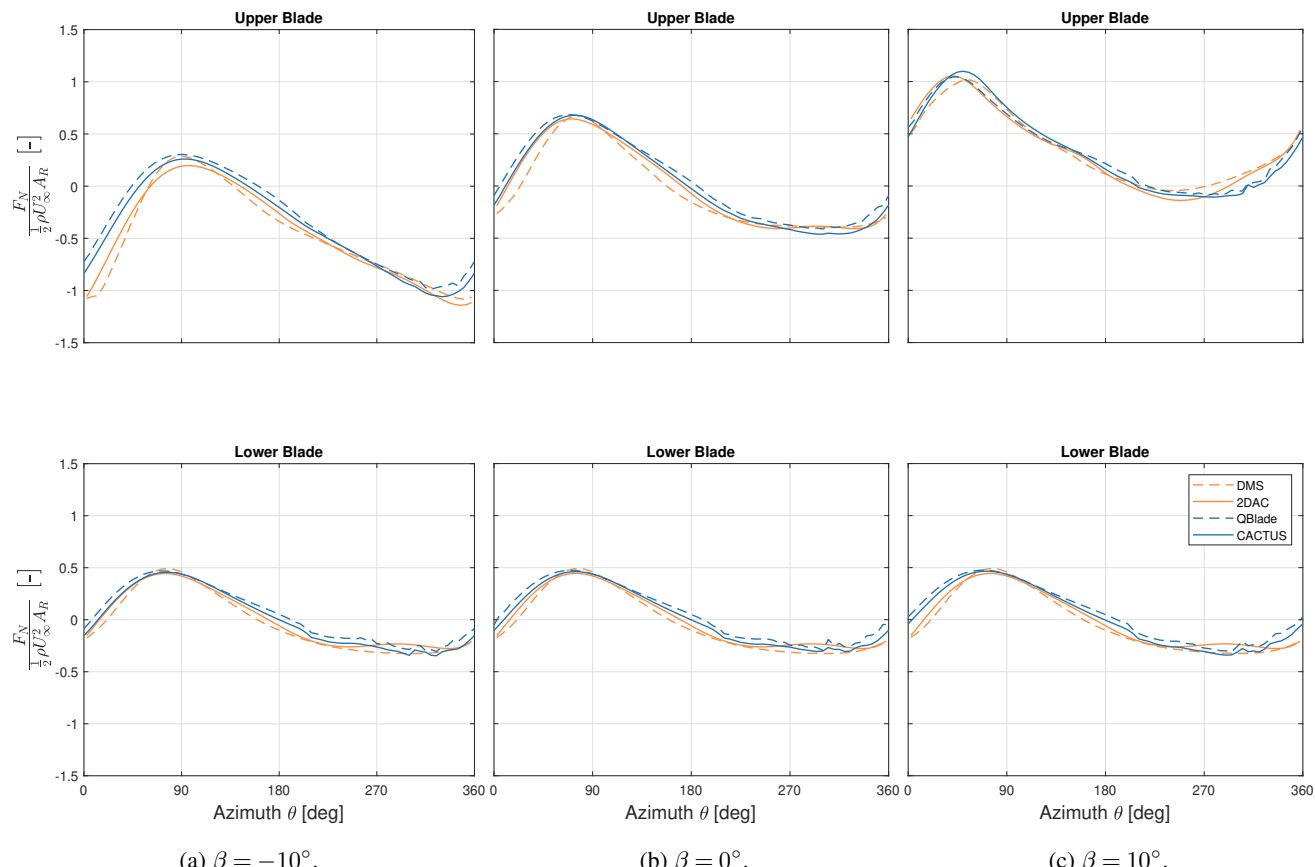

**Figure 10.** Normal force from upper & lower blades for $\beta = [-10°, 0°, 10°], \lambda = 4$. Orange and blue are the momentum & vortex models respectively.

in the first quarter of the azimuth for the upper blades. The momentum models predict larger force magnitudes than the vortex

models. This is due to the large negative angle of attack observed because of the pitch offset, causing certain points along the span to approach stall conditions. This is also observed in the last quarter. At $\beta = 10°$, the models agree well in general, but they deviate in the last azimuthal quarter for the upper blades. This difference arises from blades nearing stall conditions (as mentioned earlier) as well as some BVI being captured by the vortex models. Additionally, the rotor experiences a phase shift in the normal force as the pitch offset changes. The peak at $\beta = 0°$ is observed before $\theta = 90°$, while the peak for $\beta = -10°$

is around $\theta = 90°$, and the peak for $\beta = 10°$ is observed around $\theta = 60°$. This occurs as the blades experience stall earlier in the azimuth with positive pitch offsets and the opposite for negative pitch offsets compared to $\beta = 0°$. This phase shift is not observed in the lower blade forces by the momentum models, as they are 2D models that do not take vertical induction into account.





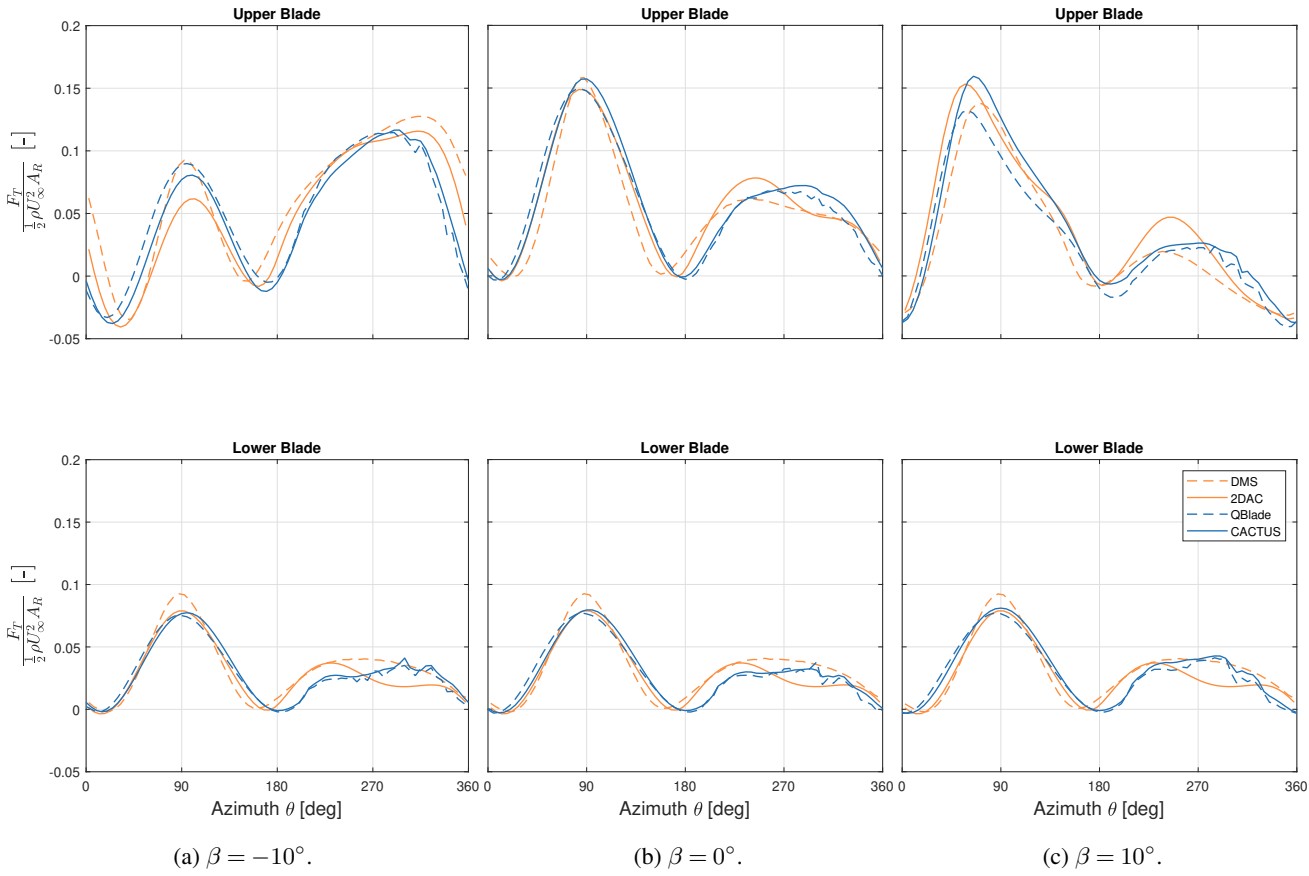

**Figure 11.** Tangential force from upper and lower blades for $\beta = [-10°, 0°, 10°], \lambda = 4$. Orange and blue are the momentum and vortex models respectively.

The tangential forces from Figure 11 also show the redistribution of forces to the upwind and downwind half with positive and negative pitch offsets respectively. For $\beta = 0°$, the models behave similarly to the trends previously seen with the normal forces in the upwind half but vary severely in the downwind half where BVI exists. This effect is enhanced with pitch offsets. The DMS model consistently predicts a higher peak tangential force for the lower blade compared to the other models. The phase shift of the forces exists for the upper blades but not for the lower blades. Again, the vortex models show slight differences in the lower blades with pitch offset despite not pitching the lower blades. The momentum models predict identical forces for the lower blades throughout the range of pitch offsets as they are 2D models.

To enhance this difference seen in the lower blade, Figure 12 shows the normalized difference in the normal force of $\beta = 10°$ and $\beta = -10°$ with respect to $\beta = 0°$ of the vortex models. The normal forces of the lower blades with pitch offsets vary from $\beta = 0°$ by roughly 6-7% from the vortex models. This shows a strong 3D aerodynamic effect of vertical induction for the X-Rotor in its operation, which is not accounted for by the 2D momentum models. This variation increases with tip-speed ratio





or larger pitch offsets, as the loads increase. In the first azimuthal quarter, where the largest relative velocity is expected, the difference starts out at $5-6\%$ but drops down to nearly $0\%$ by $\theta = 90°$. This is due to the phase shift of the peak forces between the pitched and non-pitched case, which brings down the relative value. In the second azimuthal quarter, the difference is nearly zero due to the weakened blade forces as the blades experience stall conditions. In the third quarter, the downwind passage of the blade flips the pressure and suction side, which inverts the magnitude of the force. In the last quarter, the differences

are high due to the presence of BVI and the increase in force magnitude. This is shown by the $\beta = -10°$ and $\beta = 10°$ plots crossing each other around $\theta = 300°$, where a sudden spike in the difference occurs and continues to diverge steadily beyond that. Additionally, CACTUS results show some jumps in the normal forces (notable ones at $\theta = 300°$, $330°$, and $340°$) that are due to BVI. However, QBlade shows large spikes in the entire downwind half of the rotor in contrast to the gentler spikes from CACTUS that are observed only in the last quarter. This can be attributed to the QBlade's turbulence vortex viscosity factor,

which was introduced by Leishman (2006) to account for increased vortex diffusion at highly turbulent flows. In QBlade, this affects the vortex core size at each time step, resulting in larger spikes.

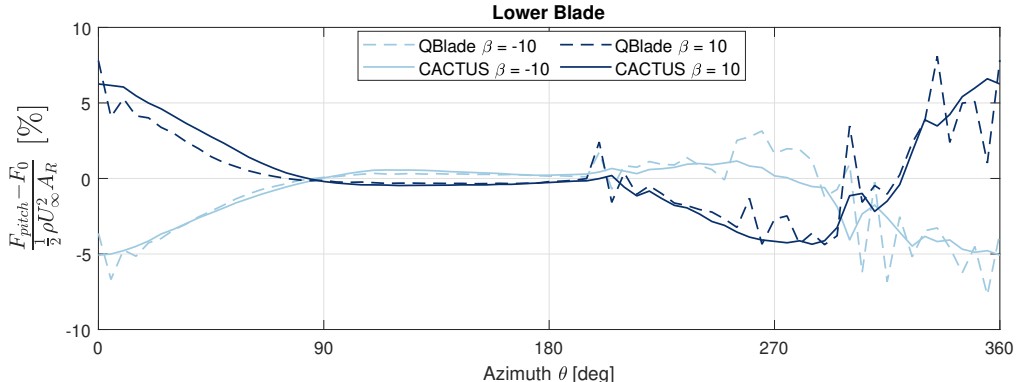

**Figure 12.** Percentage difference of normal forces at fixed pitch-offsets relative to zero pitch offset of vortex models. Light and dark blue indicate $\beta = -10°$ and $\beta = 10°$ respectively.

### 3.3 Vertical induction and inflow

The X-Rotor consists of coned blades in its primary rotor geometry, therefore a component of the normal forces from the blade acts in the axial or vertical direction. As observed in Figure 12, the presence of vertical induction while pitching the upper

blades significantly affects the forces in the fixed-pitch lower blades due to the change in force field towards the upwind and the downwind halves. CACTUS results for the vertical induction at mid-span of the X-Rotor blades along its azimuthal cycle are shown in Figure 13. QBlade results are not considered as it does not store induction in all directions in its output. The vertical induction due to coned blades as well as the finite blade effects can be observed clearly at $\beta = 0°$ for the upper blades, where the induction is mostly negative for the upwind half and mostly positive for the negative half, which is due to the blade tip

vortices as well as the coned blades of the X-Rotor. Furthermore, $\beta = -10°$ is shown to be exhibiting positive induction overall



in contrast to the negative induction shown in $\beta = 10°$. The direction of vertical induction indicates that the tip-vortices also flip their direction, which causes a large change in the flowfield of the rotor. The lower blades clearly experience a difference in vertical induction as the pitch of the upper blade changes, which correlates well with the trends that are seen in Figure 12. Most differences can be observed in the first, third, and last azimuthal quarters, while the second quarter shows the least difference

due to stalling of the blade. Furthermore, this shows the importance of vertical induction especially with blade pitch, where the force field changes the tip-vortex strength resulting in enhanced 3-dimensionality of results (Huang et al., 2023).

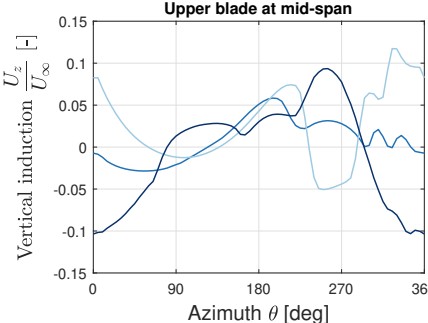
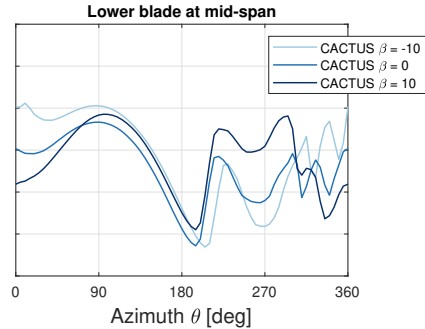

**Figure 13.** Vertical induction at mid-span of upper and lower blades. Vertical induction results are from CACTUS for $\beta = [-10°, 0°, 10°]$ indicated by light, medium, and dark blue profiles. Results are normalised by free-stream velocity.

As the tip-speed ratio is different along the span, the vertical induction also varies along the span. To obtain an overview of the spanwise variation of vertical induction at these pitch cases, Figure 14 shows contour plots of vertical induction as a function of span and azimuth. At $\beta = 0°$, the upper blades seem to exhibit almost no vertical induction in the upwind half,

while the lower blades produce small positive induction, with the tips showing the largest induction. In the downwind half, the upper blades show positive induction and the lower blades show negative induction. At $\beta = -10°$, the upwind half is mostly dominated by positive induction from the root sections and the lower blades, with the tip-vortex of the lower blade giving negative induction at $\theta = 0°$. This is because the upper blades are loaded more in the downwind half. At $\theta = 180°$, the upper blades exhibit large positive induction, while the lower blades show mostly large negative induction. However, there exists a

strong negative induction in the root region around $\theta = 300°$ before transitioning to strong positive induction. Additionally, due to the downwind loading of the upper blades, the regions close to the tip until mid-span show very large positive induction. At $\beta = 10°$, there is primarily strong negative induction in the first azimuthal quarter concentrated mostly in the upper blades due to large loading in the upwind half. Whereas in the downwind half, there exists a dominant positive induction in the root region and the upper blades before transitioning to a strong negative induction in the last azimuthal quarter. Interestingly, it is

observed that with pitch offsets, the root section is quite important despite a smaller local tip-speed ratio when compared to the tip. This is attributed to the root vortices that are formed as the upper and lower blade meet there. Change in pitch affects the vorticity system of the upper blade tips as well as the roots. Ultimately, the vertical induction varies quite significantly and it is seen that the induction on the lower blades is influenced heavily by that of the upper blades as well as the tip and root vortices.





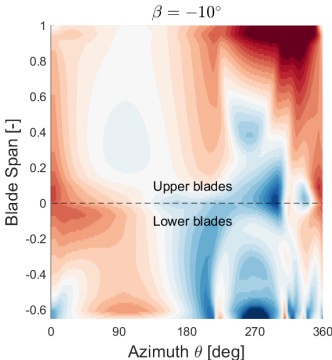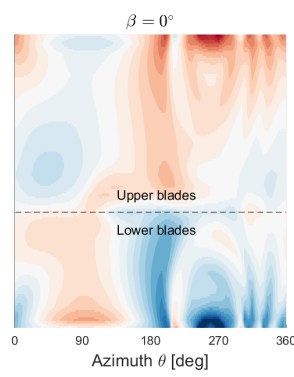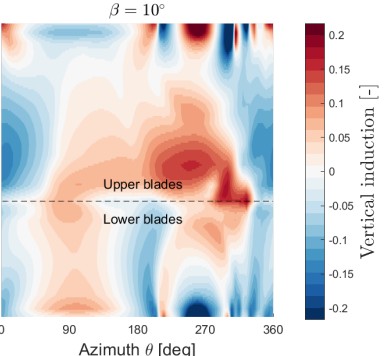

**Figure 14.** Vertical induction as a function of azimuth and blade span. Results are from CACTUS for $\beta = [-10°, 0°, 10°]$. The lower blade is represented with negative span values. Blade span is normalised with the maximum span, with 1 being the tip of the upper blades, 0 being the root, and -0.65 being the tip of the lower blades.

Overall, the force field of the rotor is subject to change with pitch offsets as the vorticity system changes (Huang et al., 2023). Therefore, it can be said that VAWTs (not specifically limited to the X-Rotor) become highly 3D with pitch offsets as the vorticity system changes and the validity of 2D aerodynamic models dwindle with larger offsets.

## 4 Conclusions

A numerical comparison of different aerodynamic models was conducted to understand the aerodynamic characteristics and the performance of an X-shaped VAWT for a range of tip-speed ratios and pitch offsets. This study contributed the following: (1) a comparative study of different aerodynamic models for the X-Rotor's primary rotor and (2) the significance of 3D aerodynamics for the X-Rotor associated with fixed blade pitch offsets.

The models presented were: Double multiple streamtube (DMS), 2D Actuator cylinder (2DAC), QBlade lifting line (QBlade), CACTUS lifting line (CACTUS), PowerFLOW (PFLOW), Unsteady Reynolds Averaged Navier-Stokes CFD model (URANS).

This study showed that the DMS model performed almost on par with the other models in calculating rotor performance parameters when there was no pitch offset, but predicted significantly different results once blade pitch was introduced, as the validity of the streamtubes enclosing the downwind actuator began to fail. This worsened at high tip-speed ratios where the DMS models predicted power coefficients at Betz limit with pitch offsets.

The 2DAC model offered consistent data at small pitch-offsets (between $\beta = [-5°, 5°]$). However, being inherently a 2D and a quasi-steady model, it did not capture the BVI and the effect of vertical induction of the X-Rotor. Moreover, it predicted the forces in the lower blade to be the same across all upper blade pitch offsets, which was not the case for the models that accounted for the vertical induction.

The thrust predicted by the momentum models was different from that of the vortex models. Moreover, these models were unable to capture the vertical induction effects and the BVI that occurred in the downwind cycle of the rotor. Therefore, the 2D





momentum models were mostly inaccurate in predicting the thrust, power, or blade loads for the X-Rotor due to the influence
of 3D aerodynamics.

The QBlade and the CACTUS open-source simulations offered great consistency with each other, including capturing the
3D aerodynamics effectively. The QBlade model showed huge turbulent viscosity spikes in its results occurring throughout
the normal and tangential forces, which was a result of the vortex core and turbulent viscosity model used in the solver.
This behaviour also translated to the blade-integrated forces where the QBlade slightly overpredicted the results compared to
CACTUS. However, both their results remained consistent, suggesting their reliability over low-fidelity models for this specific
rotor geometry.

URANS and PFLOW results were compared with the vortex and momentum models for power, thrust, and blade forces for
one test case. Primarily, the URANS and PFLOW agreed very well with the thrust of the vortex models, although predicted
slightly less power. This was because URANS had small domain sizes which caused enough blockage effects to affect the
results, and the PFLOW modelled the tower as well as the secondary rotors which also caused slight power loss. Secondly, the
downwind half of the turbine showed significantly lower force magnitudes, which was due to a combination of flow separation
at the blades as well as the inherent flow curvature effects present in the CFD solvers.

The distribution of vertical induction over the span and azimuth was studied to understand the 3D aerodynamics of the X-
Rotor, also with pitch offsets. The vertical induction was dominated primarily by the tip vortices (due to spanwise lift variation)
in cases with no pitch offset. Interestingly, with pitch offsets the root vortex showed greater influence on the vertical induction
for the X-Rotor despite operating at a much lower local tip-speed ratio. Due to the large vertical induction fluctuations through
the azimuthal cycle of VAWTs at fixed pitch offsets, it concluded that the 2D models lose their validity in these conditions and
for coned VAWTs.

*Data availability.* The generated and processed data sets used in this manuscript are available at https://doi.org/10.5281/zenodo.8208638 for
further research purposes.



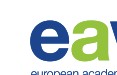
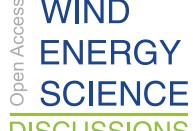

## Appendix A: Parametric definition of the blade geometry

**Table A1.** Parametric definition of the upper/top blade.

| Section | Height (m) | Chord (m) | Radius (m) | Twist (°) | Pitching axis (fraction of chord) | Foil | Reynolds number |
|---------|-----------|-----------|-----------|-----------|-----------------------------------|------|-----------------|
| 1 | 0.00 | 10.00 | 25.00 | 0.00 | 0.25 | NACA 0025 | $1.5 \times 10^7$ |
| 2 | 5.09 | 9.71 | 27.94 | 0.00 | 0.26 | NACA 0024 | $1.5 \times 10^7$ |
| 3 | 10.19 | 9.41 | 30.88 | 0.00 | 0.26 | NACA 0023 | $1.5 \times 10^7$ |
| 4 | 15.28 | 9.12 | 33.82 | 0.00 | 0.27 | NACA 0022 | $1.5 \times 10^7$ |
| 5 | 20.38 | 8.82 | 36.76 | 0.00 | 0.28 | NACA 0021 | $1.5 \times 10^7$ |
| 6 | 25.47 | 8.53 | 39.71 | 0.00 | 0.29 | NACA 0020 | $1.5 \times 10^7$ |
| 7 | 30.56 | 8.24 | 42.65 | 0.00 | 0.30 | NACA 0019 | $1.5 \times 10^7$ |
| 8 | 35.66 | 7.94 | 45.59 | 0.00 | 0.31 | NACA 0018 | $1.5 \times 10^7$ |
| 9 | 40.75 | 7.65 | 48.53 | 0.00 | 0.32 | NACA 0017 | $1.5 \times 10^7$ |
| 10 | 45.85 | 7.35 | 51.47 | 0.00 | 0.33 | NACA 0016 | $1.5 \times 10^7$ |
| 11 | 50.94 | 7.06 | 54.41 | 0.00 | 0.34 | NACA 0015 | $1.5 \times 10^7$ |
| 12 | 56.04 | 6.76 | 57.35 | 0.00 | 0.36 | NACA 0014 | $1.5 \times 10^7$ |
| 13 | 61.13 | 6.47 | 60.29 | 0.00 | 0.37 | NACA 0013 | $1.5 \times 10^7$ |
| 14 | 66.22 | 6.18 | 63.24 | 0.00 | 0.39 | NACA 0012 | $1.5 \times 10^7$ |
| 15 | 71.32 | 5.88 | 66.18 | 0.00 | 0.40 | NACA 0011 | $1.5 \times 10^7$ |
| 16 | 76.41 | 5.59 | 69.12 | 0.00 | 0.42 | NACA 0010 | $1.5 \times 10^7$ |
| 17 | 81.51 | 5.29 | 72.06 | 0.00 | 0.47 | NACA 0009 | $1.5 \times 10^7$ |
| 18 | 86.60 | 5.00 | 75.00 | 0.00 | 0.50 | NACA 0008 | $1.5 \times 10^7$ |



**Table A2.** Parametric definition of the lower/bottom blade.

| Section | Height (m) | Chord (m) | Radius (m) | Twist (°) | Pitching axis (fraction of chord) | Foil | Reynolds number |
|---|---|---|---|---|---|---|---|
| 1 | 0.00 | 14.00 | 25.00 | 0.00 | 0.25 | NACA 0025 | $1.5 \times 10^7$ |
| 2 | 2.48 | 13.59 | 27.94 | 0.00 | 0.26 | NACA 0024 | $1.5 \times 10^7$ |
| 3 | 4.95 | 13.18 | 30.88 | 0.00 | 0.26 | NACA 0023 | $1.5 \times 10^7$ |
| 4 | 7.43 | 12.76 | 33.82 | 0.00 | 0.27 | NACA 0022 | $1.5 \times 10^7$ |
| 5 | 9.91 | 12.35 | 36.76 | 0.00 | 0.28 | NACA 0021 | $1.5 \times 10^7$ |
| 6 | 12.38 | 11.94 | 39.71 | 0.00 | 0.29 | NACA 0020 | $1.5 \times 10^7$ |
| 7 | 14.86 | 11.53 | 42.65 | 0.00 | 0.30 | NACA 0019 | $1.5 \times 10^7$ |
| 8 | 17.34 | 11.12 | 45.59 | 0.00 | 0.31 | NACA 0018 | $1.5 \times 10^7$ |
| 9 | 19.81 | 10.71 | 48.53 | 0.00 | 0.32 | NACA 0017 | $1.5 \times 10^7$ |
| 10 | 22.29 | 10.29 | 51.47 | 0.00 | 0.33 | NACA 0016 | $1.5 \times 10^7$ |
| 11 | 24.76 | 9.88 | 54.41 | 0.00 | 0.34 | NACA 0015 | $1.5 \times 10^7$ |
| 12 | 27.24 | 9.47 | 57.35 | 0.00 | 0.36 | NACA 0014 | $1.5 \times 10^7$ |
| 13 | 29.72 | 9.06 | 60.29 | 0.00 | 0.37 | NACA 0013 | $1.5 \times 10^7$ |
| 14 | 32.19 | 8.65 | 63.24 | 0.00 | 0.39 | NACA 0012 | $1.5 \times 10^7$ |
| 15 | 34.67 | 8.24 | 66.18 | 0.00 | 0.40 | NACA 0011 | $1.5 \times 10^7$ |
| 16 | 37.15 | 7.82 | 69.12 | 0.00 | 0.42 | NACA 0010 | $1.5 \times 10^7$ |
| 17 | 39.62 | 7.41 | 72.06 | 0.00 | 0.47 | NACA 0009 | $1.5 \times 10^7$ |
| 18 | 42.10 | 7.00 | 75.00 | 0.00 | 0.50 | NACA 0008 | $1.5 \times 10^7$ |





## Appendix B: URANS CFD Setup

Number of cells: 72 million

The mesh background is created using blockMesh. The mesh dimensions are:

**Table B1.** Simulation domain description

|  | Minimum [m] | Maximum [m] | Number of cells | Element size with no refinement [m] |
|---|---|---|---|---|
| X direction | -400 | 840 | 161 | 7.7 |
| Y direction | -400 | 400 | 104 | 7.7 |
| Z direction | -300 | 300 | 78 | 7.7 |

The mesh has been created using snappyHexMesh, where three refinement regions have been used:

- Annular disk in the upper blades. Axis (0 0 70) to (0 0 100), radius 90 with inner radius 60.

- Annular disk in the lower blades Axis (0 0 -55) to (0 0 -35), radius 90 with inner radius 60

- Cylinder Axis (0 0 -150) to (0 0 150) radius 200m

With regard to meshing levels, the rotor surface is meshed with a level 8 that leads to a minimum element size of: 0.96 m.
The two annular disks are meshed with a level 5 which corresponds to a minimum element size of: 1.54 m. The cylinder is
meshed with a level 3 with a minimum element size of: 2.56 m. To account for the boundary layer along the blades, the rotor
is meshed with an expansion ratio of 1.2, with a minimum thickness of $1 \times 10^{-8}$ (to capture the boundary layer, this thickness
was necessary to maintain a Y+ to be less than 1), and with 5 surface layers.

The rotation of the turbine is imposed in the domain using the arbitrary mesh interface (AMI) capability available in Open-
FOAM. The boundary condition of the domain is listed in Table B2.

**Table B2.** Boundary conditions

| Boundary | Location [m] | Type [m] | Velocity condition | Pressure condition |
|---|---|---|---|---|
| XROTOR primary | N/A | Wall | movingWallVelocity | zeroGradient |
| Inlet | X = -400 | Patch | freeStream | freeStreamPressure |
| Outlet | X = 840 | Patch | freeStream | freeStreamPressure |
| Sides | Y = ±840 | N/A | freeStream | freeStreamPressure |
| Bottom | Z = -300 | Wall | freeStream | freeStreamPressure |
| Top | Z = 300 | Patch | freeStream | freeStreamPressure |

The turbulence model used in the simulations is k-$\omega$-SST. The baseline setup used the Euler discretization scheme for time
and Gauss linear schemes for the gradients. Divergence schemes are Gauss linear Upwind for velocity, Gauss upwind for k, and



$\omega$. The Laplacian schemes used are Gauss linear limited corrected. Pressure-velocity coupling is achieved with the PIMPLE equations with 3 outer corrector routines. The under-relaxation factors for velocity and pressure are 0.9.

**Appendix C: PowerFLOW Setup**

Resolution, deltax, Fine equivalent cells Physical time step, second Coarse $9.525 \times 10\text{-}3$ 7.7 million $1.439 \times 10\text{-}5$ Fine $4.76 \times 10\text{-}3$ 23.5 million $7.1942 \times 10\text{-}6$

2.1 Grid information and aerodynamic results comparison The computational setup for the finer grid resolution remains consistent with the previous report, with only changes being made to the grid resolution and physical time step which shown

in the table below. The finest cell size is $4.76 \times 10^{-3}$, which corresponds to 50 cells per average chord of the secondary rotor blade. The total number of cells and surface elements for this simulation is 291.6 million and 32.3 million respectively. The flow simulation time is 67.63 seconds which is equivalent to 9 primary rotor revolutions requiring 18995 CPU hours per revolution using Linux Xeon® Gold 6148 (twenty-core Skylake) 2.4GHz Platform.

*Author contributions.* AGA and LM did the main research work, performed and analysed the numerical results from the momentum and

vortex models, and wrote the paper with AGA writing the most. YW developed and provided the numerical results for PowerFLOW. DB, AC, and OP performed the URANS simulations and presented the results. Through meetings and feedback, AGA, LM, YW, and CF contributed to interpreting and discussing the results. CF offered general guidance and also reviewed the paper. The paper was revised and improved by all authors.

*Competing interests.* The contact author has declared that none of the authors have any competing interests.

*Acknowledgements.* The authors gratefully acknowledge the computational resources provided on the DelftBlue cluster by the Delft High Performance Computing Centre (DHPC) at the Delft University of Technology to perform the required simulations. Moreover, the authors acknowledge the support provided by William Leithead and James Carroll from the University of Strathclyde for this study.

*Financial support.* This research has been supported by the funding received from the European Union's Horizon 2020 research and innovation program under grant agreement No 101007135 as part of the project - XROTOR.





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
