# Peer review of "Aerodynamic model comparison for an X-shaped vertical-axis wind turbine"

_Wind Energy Science, 2023_

## Author Comment (AC1)

**Manuscript ID: WES-2023-115**

**Aerodynamic model comparison for an X-shaped vertical-axis wind turbine**

**January 2024**

The authors would like to thank the reviewers for their time, and valuable comments. The comments of both reviewers are useful as they improve the quality of the paper. The comments of each reviewer are addressed separately. The explanation for each question/comment is marked in blue while the actual changes in the manuscript are marked in red.

**Reviewer 1**

**Section 2.1.6**

*PFLOW simulation includes the secondary rotors. The size of the secondary rotor should be stated in the paper. In addition, it is better that information, such as the rotational speed of the secondary rotor blades and its thrust force (drag force) in the PFLOW calculation, is described in the paper. The reviewer suspects that the aerodynamic loss of the entire X-Rotor may be quite large, because the secondary rotors generate power and the propeller blades generate large drag.*

We have now added additional information about the secondary rotor in *Appendix C: PowerFLOW Setup* as we believe adding secondary rotor specification and its thrust data to the results section would clutter the main findings of the paper. The normal and tangential loads shown in the paper are only the loads of the primary rotor blades. The lower blade loads from PFLOW simulations in Figure 9 are lesser than the other models in the upwind due to the forcefield created by the secondary rotors. The design of the secondary rotors is to balance the torque on the primary rotor, which should not create huge losses in power as suspected by Reviewer 1. Further information about this is included in the paper. Line 21 has been edited to help clarify the purpose of the secondary rotors in the X-Rotor concept as: Rather than removing power from the main shaft of the primary rotor, the rotor speed is controlled by the thrust force on the secondary rotors, and all electrical power is extracted from the secondary rotors. A thrust profile of the secondary rotor in its azimuthal cycle from PFLOW is also added to *Appendix C*. The force has been non-dimensioned using the same constant for the primary rotor forces. The magnitudes that can be observed show that the influence of thrust on the primary rotor blades is very minimal, owing to also what is explained previously. The following content has been added to *Appendix C: PowerFLOW Setup*:
The simulations conducted are of the primary rotor coupled with the secondary rotor and the tower. The dimensions and the operating condition of the secondary rotor is mentioned in Table TABLE. Additionally, the thrust profile for one of the two secondary rotors for the last revolution from the PFLOW calculations is provided in Figure 1.

Table 1: Secondary rotor specifications

| | |
|---|---|
| Number of blades | 5 |
| Rotor diameter $(m)$ | 9.4 |
| Max $C_p$ | 0.27 |
| Rotational velocity $(rad/s)$ | 43.275 |
| Center effective wind speed $(m/s)$ | 61.2 |

[Figure]

Figure 1: Thrust of the secondary rotor over the azimuthal cycle of the primary rotor non-dimensionalised by the inflow velocity $U_\infty$ and frontal area $A_R$ of the primary rotor over the last azimuthal cycle of the PFLOW simulation. As the secondary rotor changes orientation at each azimuth, the thrust presented at the top is the drag force of the rotor against the freestream direction of the primary rotor and the bottom figure is the thrust acting along the rotation axis of the secondary rotor.

**Section 3.1.1**

*line 189-190: In DMS, when the pitch angle $\beta$ becomes negative, the $C_p$ increases, unlike other models. The authors think this can be attributed to the assumption of DMS in which the downstream actuator is placed in the upwind actuator's completely expanded (decelerated) flow field. The reviewer would like to request the authors to show and compare the flow field in the mid of the rotor between different models. The velocity distribution of DMS which will be compared can be the completely expanded flow speed after the upwind half.*

The authors agree with the reviewer that the paper would benefit from a further explanation of the breakdown of the DMS method. However, as the velocity field is not well defined at the mid-point of the rotor in the DMS method, the authors believe that flow field visualisation at the rotor mid-point may not be the best means of comparing the methods. We believe comparing the thrust and torque of the negative pitch case can give some insights that can explain this anomaly. In the DMS method, the flow retardation is under predicted in the downwind rotor section leading to a larger (in absolute terms) flow angles, shown in Figure 2. This is compounded by the negative thrust values in the upwind rotor section accelerating the flow, also shown in figure Figure 2. The increased flow angle leads to an increased angle of attack in downwind rotor section and increases the torque production. Additionally, through ignoring the effect of the downwind actuator surface on the upwind actuator, a larger inflow angle is also predicted in the upwind rotor portion. This leads to a positive angle of attack in the upwind section (with the inflow angle larger than 10 degrees) and positive torque production in the upwind rotor section which is not predicted by the 2D actuator cylinder model. Secondly, exaggerated streamtube expansion/contraction is demonstrated by the size of the region in which the flow angle is positive, also shown in Figure 2. This increases the size of the downwind rotor section and the range over which positive torque is produced, increasing the power coefficient estimate. The opposite occurs at positive pitch offsets, where the region in which positive torque production occurs (the upwind rotor section) in contracted. This effect explains the drastic drop in power observed at $\beta \geq 5°$. The text from line 190 is revised to read: There are two key features which lead the

breakdown of the DMS method in situations with pitch offsets. Firstly the DMS method does not consider the induction of the downwind portion of the rotor at the upwind actuator surface which becomes more critical as the loading is skewed to the downwind section at negative pitch offsets. Additionally, the streamtube expansion correction used in the DMS model over-predicts the contraction at the upwind actuator surface and the expansion at the downwind actuator surface. This leads to the region over which negative torque is produced to be contracted and the region over which the positive torque is produced to be extended leading to an over-prediction in power production in the case of negative pitch offsets. The Section 3.1.2 is also altered to reflect this detail that was left out in the previous version of the paper: The DMS model under predicts the thrust values at high loading, indicating that the correction used for high blockage cases is inaccurate. Additionally, the dependence on pitch angle does not agree with the other models. This can again be attributed to the effects of on over prediction of streamtube expansion increasing the estimate of thrust as the loading is shifted to the downwind rotor half, and decreasing the estimate of thrust as the loading is redistributed to the upwind rotor half in a similar manner to the power as discussed in section Section 3.1.1.

[Figure]

Figure 2: Inflow angle at 50% span, nondimensional torque and thrust of the upper blade at $\beta = -10°$

*line 197-199: The authors attribute the reason, why the $C_p$ is smaller in URANS compared to the vortex method, to the blockage due to the small size of the calculation area. However, if the distance of rotor from the inlet is small, the $C_p$ generally increases compared to the calculation with a sufficiently large distance. Although the authors mentioned it again in lines 338-340 of Chap 4 "Conclusions", it does not seem to be reasonable to explain that the $C_p$ of URANS is smaller than that of the vortex method because the calculation area is small.*

The authors agree that the reviewer is correct. We thought the outlet distance could be a bigger influence, as the $C_p$ is shown to reduce if the outlet distance from the rotor is lesser than optimal in Rezaeiha et al., 2017. In hindsight, as the difference in $C_p$ prediction of the URANS and the vortex models is only about 6%, the authors believe this small difference is due to the discrepancy in lift and drag between airfoil polars used in vortex models and the blade-resolved computations from URANS instead of the previous reasoning. This is also explained by the comparison of XFOIL with other numerical and experimental methods at different Reynolds numbers by Melani, P. F., Balduzzi, F., Ferrara, G., & Bianchini, A. (2019), An Annotated Database of Low Reynolds Aerodynamic Coefficients for the NACA0021 Airfoil. AIP Conf. Proc, 2191, 20111. https://doi.org/10.1063/1.5138844. The lines 198 and onwards has been modified to reflect this: While blockage effects can affect the $C_p$ from the URANS simulations (Rezaeiha et al., 2017a), the difference mainly arises from the use of polars in the vortex models versus performing a full blade-resolved study. This is demonstrated in a study conducted by Melani et. al. (2019) where they analysed the lift and drag coefficients for a NACA0021 airfoil between XFOIL polars as well as the blade resolved 2D URANS results. The results showed significant differences at low Reynold's numbers, but at higher Reynolds numbers, the differences minimise but do not vanish.

Additionally, the difference in URANS and vortex models discussed in Section 3.1.2 is altered to reflect what is mentioned above: The difference between URANS and the vortex models (around 2%) is again due to the use of polars vs blade resolved aerodynamic forces, as mentioned earlier.

Appearance etc.

The authors are unclear on where this comment points to. However, we have also identified typos and other small inconsistencies in the manuscript and have corrected them.

**Appendix C**

*line 371-372: Correct the way you write the sentence "Resolution, deltax, ....... 7.1942×10-6", please.*

This sentence has been made into a small table captioned *Grid and time-step sizes*. The "fine equivalent" values have been replaced by the actual cell count instead.

Table 2: Grid and time-step sizes

| Resolution type | $\Delta x$ (m) | Cell count | Time-step (s) |
|---|---|---|---|
| Coarse | $9.525 \times 10^{-3}$ | 43.2 million | $1.439 \times 10^{-5}$ |
| Fine | $4.76 \times 10^{-3}$ | 291.6 million | $7.1942 \times 10^{-6}$ |

*line 373 "2.1 Grid —— comparison" Check please.*

The sentence has now been removed as it was not necessary.

*line 375 4.76×10-3 → $4.76 \times 10^{-3}$ or 4.76E-3*

It is now corrected to: $4.76 \times 10^{-3}$

*line 377 18995 CPU hours per revolution → 18995 CPU hours (about 40 days) per revolution (?) Revise, please.*

There was a typo. The PFLOW simulations used 500 cores for calculation, which takes 1.5 days per revolution. The sentence has been modified: The flow simulation time is 67.63 seconds which is equivalent to 9 primary rotor revolutions requiring 18995 CPU hours per revolution using Linux Xeon® Gold 6148 (500-core Skylake) 2.4GHz Platform.

**Reviewer 2**

*line 149: Table 2 includes twist, which is not mentioned or discussed.*

We have now removed the twist column from Table 2 as it is irrelevant due to the lack of twist distribution in both blades.

*line 156: 72 azimuthal discretizations per rotation big; QBlade can be sensitive to this. Did the authors consider other discretizations?*

Yes. We performed a sensitivity study between 4 different azimuthal discretizations: 36, 60, 72, and 144 in QBlade (to get 10°, 6°, 5°, and 2.5° per time-step respectively) for $C_p$ convergence. We have attached the sensitivity results below (Figure 3). We found that between 72 and 144, there was only a 0.2% difference in $C_p$, which was considerably small. As the trend shows, finer discretizations would yield less error. However, the 144 discretization case took nearly 3 times longer than the 72 discretizations case. Furthermore, we wanted consistency between the vortex models (with the number of vortex filaments, the time-step size and the blade section discretizations) as well as the DMS and 2DAC models. Due to these factors, we chose 72 discretizations for all our simulations.

*line 227: PFLOW lower blade forces are affected by the tower blade interaction and the secondary rotors included. There is no mention of this.*

The blade tower interaction is mentioned in lines 227 and 228. We have updated it to include the discus-

[Figure]

Figure 3: Azimuthal discretization senisitivity study conducted for QBlade for 4 different discretization levels. Y axis on the left shows mean $C_p$ of last revolution from QBlade and the axis on the right shows the percentage error with respect to the power at 144 discretizations.

sion of the influence of the forcefield due to the secondary rotor on the primary rotor blades in the paper: This is due to the forcefield created by the induction of the secondary rotor mounted on the lower blades. This induction of the secondary rotor has a small area of influence which does not affect the upper blades significantly. Moreover, specifications of the secondary rotor is now added to *Appendix C: PowerFLOW setup.*